# Gender differences in *Leptospira* exposure risk, perceptions of disease severity, and high-risk behaviours in Salvador, Brazil: A cross-sectional study

Ellie A. Delight[1], Diogo César de Carvalho Santiago[2], Fabiana Almerinda G. Palma[2], Daiana de Oliveira[2,3], Fábio Neves Souza[2], Juliet Oliveira Santana[2], Arata Hidano[1], Yeimi Alzate López[2], Mitermayer G. Reis[3], Albert I. Ko[3,4], Akanksha A. Marphatia[1], Cleber Cremonese[2☯], Federico Costa[2,3,4☯*], Max T. Eyre[1☯]

**1** Department of Disease Control, London School of Hygiene & Tropical Medicine, London, United Kingdom, **2** Collective Health Institute, Federal University of Bahia, Salvador, Brazil, **3** Gonçalo Moniz Institute, Oswaldo Cruz Foundation, Salvador, Brazil, **4** Department of Epidemiology of Microbial Diseases, Yale School of Public Health, New Haven, Connecticut, United States of America

☯ Contributed equally.
* fcosta2001@gmail.com

## Abstract

Vulnerability to climate hazards and infectious diseases is not gender-neutral, meaning that men, women, and other gender identities experience different risks. Leptospirosis is a zoonotic climate-sensitive infectious disease caused by the bacteria *Leptospira* and is transmitted to humans through contact with infected animals or contaminated environments, particularly soil and floodwater. Globally, studies report a higher risk of leptospiral infection among men than women, a trend also observed in Salvador, Brazil; however, the factors driving this difference are poorly understood. This study aimed to investigate how *Leptospira* exposure differs between men and women living in urban informal settlements in Salvador. We conducted a cross-sectional serosurvey among 761 adults (280 men and 481 women) in four communities previously identified as high-risk by surveillance data. Using a causal inference approach and a two-part sex-disaggregated analysis, we applied logistic regression models to examine: (1) the association between perceived severity and high-risk behaviours with *Leptospira* seropositivity, and (2) the association between perceived severity with high-risk behaviours. Seroprevalence was 14.6% (95% CI: 10.5%-18.8%) in men and 9.4% (95% CI: 6.7%-12.0%) in women. Men who perceived leptospirosis as extremely serious had lower odds of being seropositive (OR: 0.38, 95% CI: 0.15-0.99), walking through sewage (OR: 0.41, 95% CI: 0.17-1.00), and walking barefoot outside (OR: 0.24, 95% CI: 0.08-0.76) compared with men who perceived leptospirosis as less serious. These associations were not observed in women and differed across age groups in men. Behaviours were not associated with seropositivity in either gender. Our results identify perceived severity as a potential

**Data availability statement:** All relevant data for this study are publicly available from the OSF repository (https://osf.io/ebdtj/).

**Funding:** This project was jointly funded by Wellcome Trust (218987/Z/19/Z; Grant recipient: FC) and the Department of Health and Social Care (DHSC) through the National Institute for Health Research (NIHR) (https://www.nihr.ac.uk/) using the UK Official Development Assistance Fund (ODA). ED was supported by a London School of Hygiene and Tropical Medicine travel grant. FC was supported by the Brazilian National Research Council (CNPq) (https://www.chistera.eu/cnpq). MTE was supported by a Reckitt Global Hygiene Institute (RGHI) fellowship (https://rghi.org/). AK was supported by the NIHR (R01 AI121207, U01 AI088752, R01 TW009504). The funders had no role in the study's design, data collection and analysis, decision to publish, or manuscript preparation.

**Competing interests:** I have read the journal's policy and the authors of this manuscript have following competing interests: MTE has received a research fellowship from Reckitts Global Hygiene Institute and a contract from Unlimit Health; FC has received grants from NIH/NIAID, Fundação de Amparo a Pesquisa do Estado da Bahia, Wellcome Trust, Oswaldo Cruz Foundation, and the Secretariat of Health Surveillance, Brazilian Ministry of Health; AIK has received grants from HIH/NIHAID and NIH/FIC, has patents issued across various Leptospira-associated proteins and sample preparation protocols, participates in Data Safety Monitoring Boards across Reckitt Global Health Hygiene Institute, National Academics of Science Engineering and Medicine, and the Global Leptospirosis Environmental Actional Network (GLEAN), World Health Organisation, and is on the Board of Directors for the American Society of Tropical Medicine and Hygiene; there are no other relationships or activities that could appear to have influenced the submitted work.

driver of high-risk behaviours and exposure in men, indicating perceptions as targets for health promotion programs, while also highlighting evidence gaps in understanding exposure risks among women. As the first sex-disaggregated study investigating *Leptospira* exposure risks, we advocate for a gendered lens in future studies to understand gender-specific risks.

## Introduction

As climate change progresses, climate-related hazards such as flooding are becoming more frequent and severe, posing significant risks to human health. A systematic review revealed that 58% of human infectious diseases are exacerbated by climatic hazards, characterised as climate-sensitive infectious diseases (CSIDs) [1]. The review highlighted that CSIDs are transmitted to humans through 1,006 unique pathways sensitive to climate hazards, with 121 of these pathways attributed to flooding. These diseases are predicted to disproportionately affect urban informal settlements which are vulnerable to the effects of flooding events due to inadequate provision of drainage systems and other basic urban services https://www.sciencedirect.com/science/article/pii/S2405880724000232.

Vulnerability to flooding and CSIDs is not gender-neutral, meaning that men, women, boys, girls, and other gender identities living in the same environments experience different health risks [2–5]. For example, men can face greater risk of exposure to CSID pathogens attributed to outdoor occupations and subsequent contact with the environment and animals, while women can face greater socioeconomic effects of CSIDs due to heightened job insecurity [6–8]. Additionally, adverse health outcomes in women are accentuated during flooding due to limited mobility or access to services, and traditional caring roles, rendering them less climate resilient [9–11].

Flooding is an important driver of leptospirosis, a zoonotic neglected CSID caused by pathogenic bacteria of the genus *Leptospira* [1,4,8,12,13]. In urban informal settlements, leptospires are shed into the environment in the urine of the infected rat reservoir [14]. Leptospires can persist for days or weeks in the environment which makes it an important reservoir and source of infection for humans through contact with soil, mud, and water [14]. Flooding exacerbates this problem by dispersing leptospires across the environment and bringing humans directly into contact with contaminated water, increasing the risk of spillover transmission at the human-animal-environment interface [4,8,15–17].

Globally, leptospirosis is more prevalent in men, who are reported as being at greater risk of *Leptospira* infection across a wide range of geographical contexts [13]. This is the case in climate-vulnerable urban informal settlements in Salvador, Brazil, where men are reported to have over twice the risk of *Leptospira* infection than women [4,8,18,15]. While the mechanisms driving this gender difference in infection rates are poorly understood [4,14,17,18], they have largely been attributed to exposures that are more prevalent in men, and there is no evidence to suggest that they are caused by physiological differences between sexes [8]. Whilst previous studies

have investigated the odds of one sex having a higher risk of exposures or *Leptospira* seropositivity compared to another sex [4,8], none have examined whether different pathways shape this risk for the two sexes (or different genders), by for example, conducting sex-disaggregated analyses of risk factors.

Notably, the disease burden in women should not be underestimated; misdiagnosis with pregnancy-related conditions in women is common and can result in maternal and foetal death [19]. Additionally, active surveillance in Salvador from 2003 – 2005 reported that while the majority of leptospirosis-associated Severe Pulmonary Haemorrhagic Syndrome (SPHS) patients were male (70%), women had almost three times the odds of developing SPHS compared to men (OR: 2.87, 95% CI: 1.36, 5.98) [20].

While research employing a gender perspective to the study of leptospirosis remains scarce, several studies in Salvador have suggested that behaviour may be an important determinant of the gender-specific risks associated with the disease [8,21,22]. For example, a previous study used GPS to monitor residents' mobility and found that men covered a larger area than women over a 24-hour period, as did infected male participants when compared with uninfected males [22]. However, exposure to environmental sources of leptospirosis transmission did not vary by gender, although this may have been due to challenges in accurately measuring environmental exposure [22]. A knowledge, attitudes, and practices (KAP) study, also conducted in Salvador, found that men had lower leptospirosis knowledge scores compared to women, which was associated with reduced protective hygiene-related behaviours when in contact with the environment, such as wearing gloves when handling waste [8]. Upon adjusting for KAP, male gender was no longer a risk factor for *Leptospira* infection, suggesting that the higher rates in men could be attributed to differences in knowledge and attitudes and the subsequent impact on behaviours. This finding aligns to the Rational Model Theory [23] that behaviour is determined by knowledge and perceptions of disease [24–26]. Collectively, these findings suggest that a complex interplay between social norms, knowledge, perceptions, and behaviours may result in varying levels of exposure to the contaminated environment, leading to differing *Leptospira* infection risk among genders.

The use of causal inference methodology and application of sex- or gender-disaggregated frameworks are important for effective study of gender-determined health risk. Causal inference methodology, utilising directed acyclic graphs (DAGs), enables researchers to map out causal systems and assumptions, offering a visual representation of complex relationships including those related to gender-determined risk [27]. Disaggregating epidemiological analyses by sex and gender, as recommended in the WHO toolkit for incorporating intersectional gender analysis into research on infectious diseases of poverty [28], allows for a gender-sensitive focused examination of infection risks. Understanding the underlying drivers of exposure to the environment for men and women is increasingly important as climate change continues to expose marginalised individuals to CSIDs and embed gendered inequalities [1–3,5].

The aim of this study was to understand how *Leptospira* exposure risk varies among men and women living in four informal settlements in Salvador, Brazil, by considering the role of perceptions and behaviours within a sex-disaggregated framework. First, we investigate the relationship between sociodemographic factors, perceptions, and behaviours with *Leptospira* infection in men and women. Then, we investigate how perceived severity of leptospirosis impacts behaviours in men and women. Finally, a sub-analysis was conducted to investigate the association of perceived severity with high-risk behaviours across age groups in men and women.

## Materials and methods

### Study design

We conducted a cross-sectional study in the city of Salvador (estimated population 2,417,678 in 2022 [29]), north-eastern Brazil. Data were collected from 16th September 2021–12th December 2022.

**Study area.** The study was conducted across four informal settlement communities in Salvador: Nova Sussuarana (NS), Arenoso, Jardim Santo Inácio/Mata Escura (JSI/ME), and Calabetão (Fig 1A). The communities were identified as high-risk for *Leptospira* by surveillance data collected between 1996–2018 using the Notifiable Diseases Information

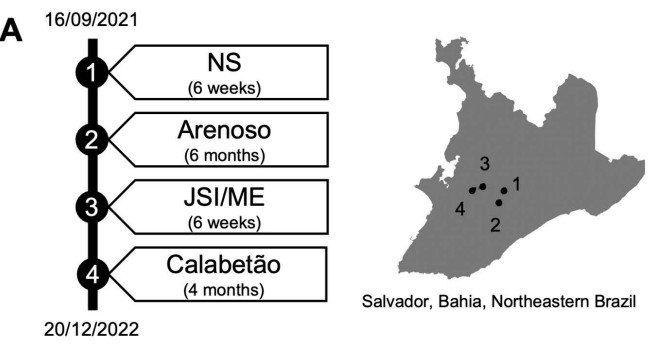

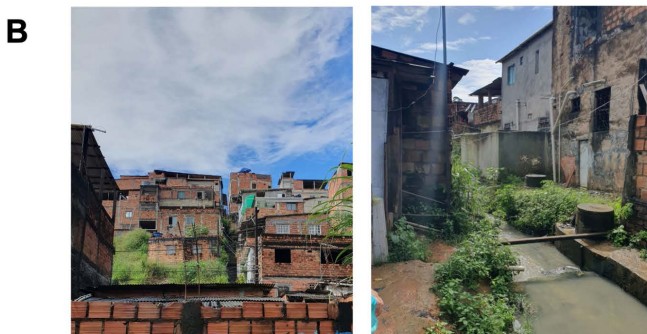

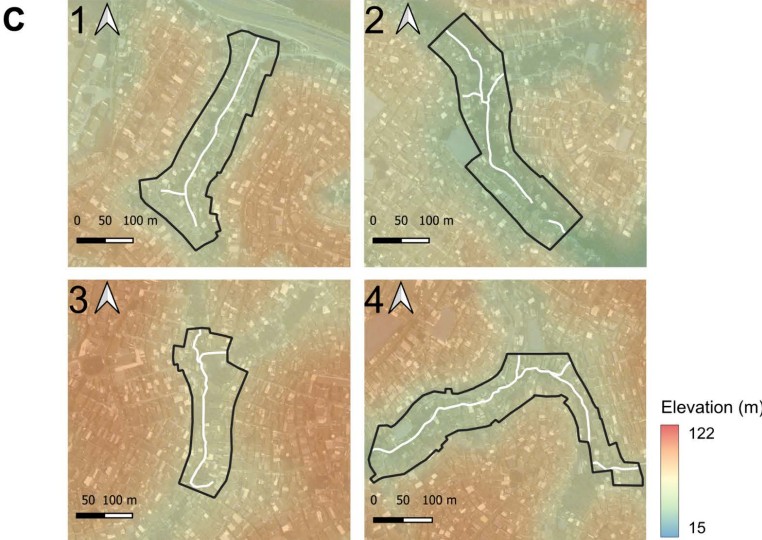

**Fig 1. Study site and timeline.** A) Cross-sectional serosurvey timeline and location of communities in Salvador, Brazil shown as numbered points. NS: Nova Sussuarana; JSI/ME: Jardim Santo Inácio/Mata Escura; B) Study site photos from Nova Sussuarana; C) Study sites shown as polygons with elevation and the position of open sewer indicated with white line. Salvador boarder sourced from GADM [33] and satellite imagery from WorldView-3 on 15/04/2022 [34].

System (SINAN) [30,31]. All communities are characterised as densely populated urban informal settlements, where residents face inadequate sanitation, drainage, and waste management systems (Fig 1B). According to the World Bank's poverty threshold [32], poverty levels are high across the communities, with the average monthly income per capita across communities (R$463) being less than half the Brazilian minimum wage in 2022 (R$1,212) [32].

The study site in each community was defined as an area within 40 metres of an open sewer (Fig 1C), identified through a previously conducted census [30]. The census included all houses located within 40 meters of the open sewer located within each study area, along a 150 – 200 metre stretch. This radius ensured the inclusion of residents living in areas with a higher risk of leptospirosis transmission. The total census counted 1,891 residents living in households within the four areas. From this population, individuals were invited to join the study if they had lived in ground floor households for at least six months, had slept at the property for at least 3 nights in the previous week, and were aged 18 years or older [30]. A minimum period of six months was established as the threshold to consider an individual as having resided in the area for long enough to accurately represent the epidemiological characteristics of the study population. Written informed consent was obtained from each participating individual; consent forms are securely stored at the Federal University of Bahia and will be kept for up to 10 years.

**Questionnaires and serosurveys.** Serosurveys were conducted within the study sites over a 15-month period. The duration of data collection in sites differed (Fig 1A) due to adverse weather and safety challenges. Data were collected during mornings from Monday – Sunday in attempt to include both employed and unemployed participants and mitigate selection bias. Trained phlebotomists visited participant homes, collected blood samples, and conducted a modified version of a standardised questionnaire that has been validated previously [4,15]. Houses were revisited up to five times if participants were not present at the time of the previous visit. Data about sociodemographic indicators (age, sex, race, highest level of school studied, employment status, occupation), perceptions, and behaviours that relate to exposure were collected. Sex and race were self-reported. Highest level of school studied was defined as either primary school (including up to 9th grade) or secondary school (above 9th grade). Employment status was defined as whether a participant had been employed during the previous week. High-risk occupations were defined as construction, street sweepers or vendors, recycling collectors, or cleaning services. Most behaviours (walking through flood water, sewage water, mud, or walking barefoot outside of the home) were characterised as the frequency at which an individual had performed a given behaviour in the previous six months (never, rarely, sometimes, often, always). Participants were also asked if they were able to wear boots during flooding in the previous six months (owned boots, could borrow boots, or couldn't access boots). Perceived severity was ranked on a five-point scale from not serious to extremely serious. Data were collected and securely stored using Research Electronic Data Capture (REDCap) electronic data collection tools hosted at Oswaldo Cruz Foundation (Fiocruz), Bahia [35,36].

A microscopic agglutination test (MAT), the reference assay of sero-diagnosis of leptospirosis, was used to measure titres of agglutinating *Leptospira* antibodies [37]. Serological samples were tested for *Leptospira* strains: *L. kirschneri* serogroups Cynopteri serovar Cynopteri strain 3522C and Grippothyphosa serovar Grippothyphosa strain Duyster; *L. interrogans* serogroups Canicola serovar Canicola strain H. Utrecht IV and Autumnalis serovar Autumnalis strain Akiyami A; and *L. borgpetersenii* serogroup Ballum serovar Ballum strain MUS 127. The panel also included two local clinical isolates: *L. interrogans* serogroup Icterohaemorrhagiae serovar Copenhageni strain Fiocruz L1-130 and *L. santarosai* serogroup Shermani strain LV3954. Individuals were classified as seropositive if their serological sample returned an MAT titre of ≥1:50 for any serogroup. Laboratory testing was conducted in the Laboratory Pathology and Molecular Biology at Fiocruz, Salvador.

## Causal inference framework

A causal inference approach was used to investigate how *Leptospira* infection risk varies by gender with language used herein following that outlined by Tennant et al., 2021 [27]. A DAG was used to map the causal hypotheses underlying

this study and to identify a sufficient set of adjustment variables in subsequent regression models [27]. The DAG was created on Dagitty [38] and a simplified version is shown in Fig 2 (the full version can be found at https://dagitty.net/dags.html?id=XxPTXytr or S1 Fig). For the development of the DAG, causal relationships between exposures and seropositivity were considered using plausible causal assumptions and assessed under temporality and biological plausibility of the Bradford Hill criterion [39]. Our key causal assumption, as mapped in Fig 2, is that gender is a distal causal factor that influences risk of *Leptospira* seropositivity. We hypothesise that gender acts directly on seropositivity, through unmeasured variables currently unknown, and indirectly through perceptions of disease, behaviour, and household environments.

## Statistical analysis

Study participation was assessed using the STROBE (STrengthening the Reporting of OBservational studies in Epidemiology) statement for cross-sectional studies [40]. Chi-squared tests were used to compare non-participation rates across sociodemographic indicators available among both those that agreed and disagreed to participate (neighbourhood, age, and sex).

A sex-disaggregated analysis was conducted to identify gender-determined risk factors for *Leptospira* infection. The analysis was conducted in two parts (Fig 3) to estimate the total causal effect of: 1) sociodemographic, perception, and behaviour variables on *Leptospira* seropositivity, and 2) perceptions on behaviours. Due to data availability, our study investigates the genders of men and women and uses these terms interchangeably with male and female based on the UNICEF definition of gender as "differences by sex and the unique needs of males and females [which] reflect differences by gender, the socially and culturally constructed roles, responsibilities, and expectations of men and women, and boys and girls" [41].

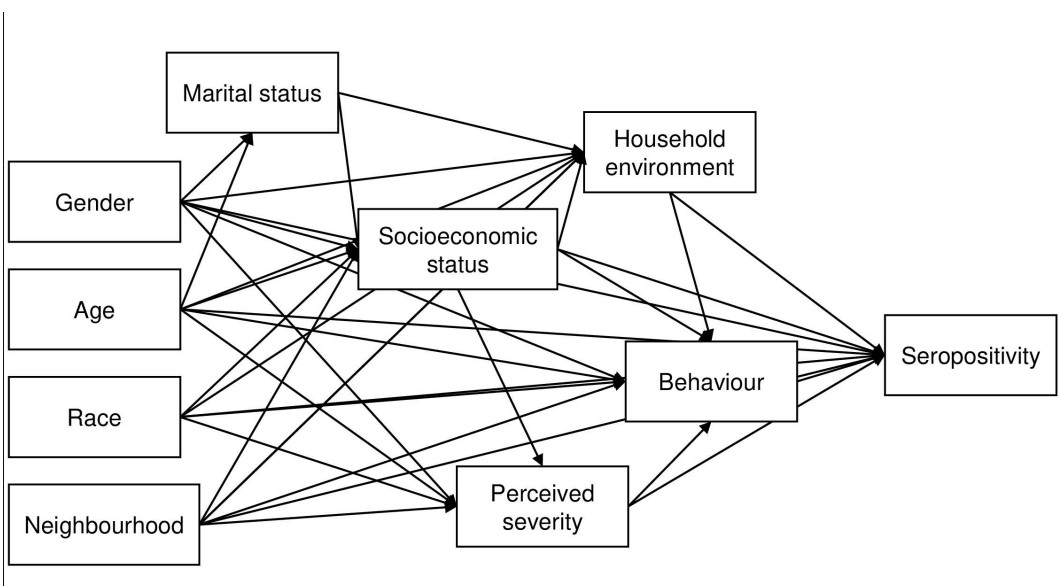

**Fig 2. Simplified DAG, where direction of causality is indicated by arrows.** Socioeconomic status: household food insecurity, education, employment, occupation; Household environment: flooding in household, living within 10m of open sewers of waste accumulation, and count of rats sighted around the household; Behaviour: Walking through sewage water, mud, or floodwater, ownership of boots, or walking barefoot.

PLOS Global Public Health

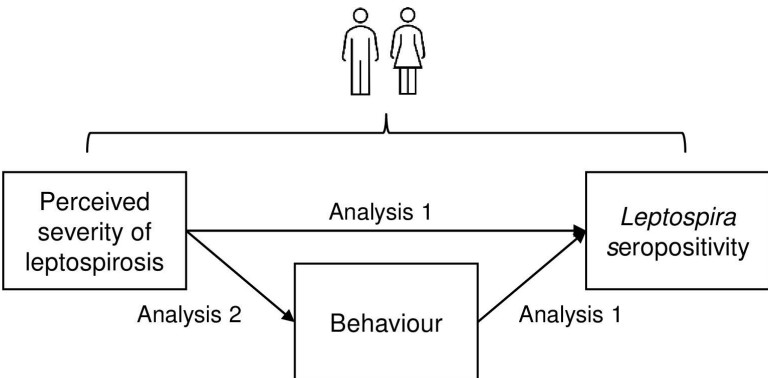

**Fig 3. Hypothesised causal pathway of how perceived severity of leptospirosis causes *Leptospira* seropositivity in men and women.** Analysis 1: Seropositivity risk actor analysis; Analysis 2: Perceived severity as a determinant of high-risk behaviours.

## Analysis 1: Seropositivity risk factor analysis

In Analysis 1 we estimated the total causal effect of each sociodemographic, perception, and behaviour variable (exposures) on *Leptospira* seropositivity (outcome) in men and women.

**Descriptive.** Descriptive cross tabulations were used to describe the prevalence of exposures across seropositivity by gender.

**Generalised additive models (GAMs) and regression analyses.** GAMs were used to assess linearity and identify the functional form of the univariable relationship between age, a continuous explanatory variable, and seropositivity [42]. In each of the multivariable regression models in Analysis 1, age was modelled as a continuous variable based on the functional form identified in the GAM analysis (S2 Fig).

Univariable binomial generalised linear regression mixed-effect models (GLMMs) were used to estimate the crude total causal effect of each exposure on seropositivity, including a household-level random effect to account for household clustering.

Multivariable GLMMs, also including a household-level random effect, were used to estimate the adjusted total causal effect of each exposure on seropositivity by adjusting for the sufficient adjustment sets identified in the DAG [38]. Each adjustment set was tested for multicollinearity using Cramer's V Correlation test, and no adjustment sets showed evidence of collinearity between categorical variables (Cramer's V coefficient < 0.08) [43]. Mediator-outcome confounders (defined as variables that are confounders for the relationship between a given mediator and outcome pair) were added to the minimal adjustment sets to improve estimate precision [27]. Mediator-outcome confounders differed by model but were limited to age, race, season, and community.

For both univariable and multivariable models, the effect of each exposure on seropositivity was estimated for the following sex-combined and sex-disaggregated populations: i) male and female; ii) male-restricted; iii) female-restricted. This resulted in three models per exposure. Only model coefficients for exposure variables were reported to avoid "Table 2 fallacy", whereby coefficients of confounders are displayed and mistakenly interpreted [27,44].

Finally, the strength of evidence supporting sex as an effect modifier for the relationship between each exposure and outcome pair was investigated by including an interaction term in the multivariable model.

## Analysis 2: Perceived severity as a determinant of high-risk behaviours

In analysis 2 we estimated the total causal effect of perceived severity of leptospirosis (exposure) on the risk of performing high-risk behaviours (outcome). These behaviours related to personal hygiene practices and the frequency of contact

with known sources of environmental risk: being able to wear boots during flooding, walking through flood water, sewage water, and mud, walking barefoot outside of the home.

The same modelling approach was followed for these exposure-outcome pairs as for Analysis 1, including both descriptive, GAM, and regression analyses. Categorical behaviour variables with multiple levels were recategorised for inclusion into binomial models as response variables. Behavioural variables were dichotomised into "rarely" (inclusive of "never" and "rarely" observations) and "frequent" (inclusive of "sometimes", "often", and "always" observations). The ability to wear boots during flooding was dichotomised into "yes" (inclusive of "yes, can borrow", and "yes") or "no" (inclusive of "no"). Most participants ranked leptospirosis as an extremely serious disease. Therefore, to account for the small number of observations across lower rankings, the perceived severity variable was dichotomised into extremely serious (inclusive of "extremely serious" only) and less serious (inclusive of "not serious", "a little serious", "serious" and "very serious").

Age was modelled based on the functional forms identified by the GAMs in S3 Fig. Subsequently, age was treated as a continuous variable or modelled as linear piecewise splines with knots at specific ages.

**Sub-analysis: Perceived severity and high-risk behaviours across age groups.** A sub-analysis was conducted to investigate the association of perceived severity with high-risk behaviours, identified as important from analyses 1 and 2, across age groups. A descriptive analysis was conducted to investigate the prevalence of perceived severity and high-risk behaviours by age group in men and women (S4 Table) using Fisher's exact test and Pearson's Chi-squared test. Multivariable generalised linear models (GLMs) with binomial structure were used to estimate the adjusted total causal effect of perceived severity with high-risk behaviours stratified by age group (S4 Fig), adjusting for the sufficient adjustment sets identified in the DAG. The oldest age group (>60 years) was not included in the sub analysis due to small sample sizes across both genders. GLMs were used rather than GLMMs because stratification by both age and gender resulted in smaller sample sizes within each group, and an exploratory analysis confirmed that most individuals within each stratified group belonged to different households, negating the need to adjust for household clustering effects.

Data were cleaned, treated, and analysed in R version 4.3.1 [45]. Various R packages were used including 'lme4', 'splines', and 'qgam' [15,46–50].

## Ethics

Ethical approval for this study was obtained from the CEP/CONEP ethics committee (CAAE 32361820.7.0000.5030). All participants involved in the study provided written informed consent prior to data collection.

## Inclusivity in global research

Additional information regarding the ethical, cultural, and scientific considerations specific to inclusivity in global research is included in the Supporting Information (S1 Checklist).

## Results

### Study overview

**Study participation.** Across the four communities, we identified 688 households that were eligible using a baseline community census and household visits, 657 of which contained individuals aged ≥ 18 years and were eligible for inclusion in this analysis (Fig 4) [30]. Of the 657 households, 481 (73%) head-of-households gave consent to participate, which permitted 916 residents eligible to participate. Of these 916 residents, 147 (16%) declined and 8 (1%) were excluded from this analysis due to undetermined serostatus, resulting in 761 participants included in this study. Non-participation rates were similar across communities (12%, 20%, 16%, and 16% in NS, Arenoso, JSI/ME, and Calabetão respectively, $\chi^2 = 5.4$, $df = 3$, $p = 0.1$). Individuals who declined to participate were younger on average than those who participated (38 vs. 42 years respectively, t-value = -2.6, $df = 204.7$, $p = 0.009$) and were significantly more likely to be male than female (70% vs. 37% respectively, $\chi^2 = 52.4$, $df = 1$, $p < 0.001$).

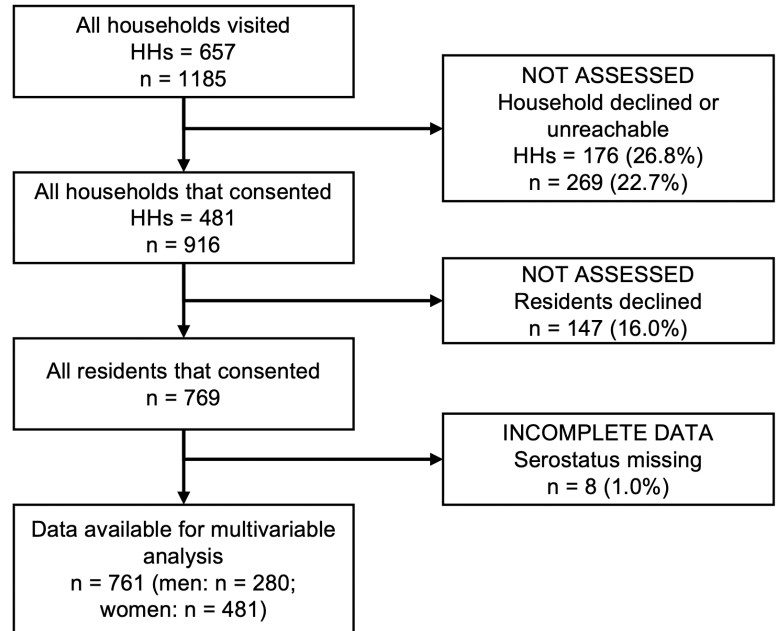

**Fig 4. Study participant flow chart aligned with the STROBE (Strengthening the Reporting of OBservational Studies in Epidemiology) statement [40]; Percentages are calculated relative to the total of the nearest previous level; HH: number of households; n: number of individuals.**

**Study population demographics, perceptions, and behaviours.** Of the 761 participants included in the analysis, 481 (63%) were female, the age ranged from 18 – 86 years, and the age distribution was similar across men and women (Table 1). The racial distribution was similar across genders, and most participants identified as Pardo (43%), indicating mixed ethnic backgrounds, or black (51%). Unemployment was higher in women than men (58% vs. 34% respectively, $\chi^2 = 40.0$, $df = 1$, $p < 0.001$).

Seroprevalence for *Leptospira*-specific antibodies was greater in men than women (15% vs. 9% respectively, $\chi^2 = 4.9$, $df = 1$, $p = 0.03$). Out of the 86 seropositive individuals, the majority were seropositive for Icterohaemorrhagiae strain Fiocruz L1-130 (69%) or Cynopteri strain 3522C (19%) serogroup. Leptospirosis was perceived as an extremely serious disease by a large proportion of both men and women (88% vs. 87% respectively, $\chi^2 = 0.1$, $df = 1$, $p = 0.8$). The proportions of men and women frequently conducting high-risk behaviours (walking through sewage water, flood water, mud, or walking barefoot outside) were similar. However, a significantly larger proportion of women lacked access to boots to wear during flooding than men (82% vs. 50% respectively, $\chi^2 = 83.1$, $df = 1$, $p < 0.001$).

The prevalence of high-risk behaviours differed by age in men and women (S3 Fig, S6 Table). In both genders, the percentage of individuals walking barefoot outside decreased linearly with age. For example, 38.0% of men aged 18–30 years walked barefoot, which reduced to 17% in those over 60 years. Similarly, 39% of women aged 18–30 years walked barefoot outside, decreasing to 6% in those over 60 years. The prevalence of walking through sewage water was also lowest among older age groups, with 13% of men and 6% of women over 60 years engaging in this behaviour. Perceived severity of disease was high across all age groups for both genders; for instance, 87% of men aged 18–30 and 73% of men over 60 reported high perceived severity, while 91% of women aged 18–30 and 85% of women over 60 reported similar concerns.

### Analysis 1: Seropositivity risk factor analysis

**Descriptive.** Seroprevalence differed by perceived severity of leptospirosis and age group in both men and women (Fig 5). Seroprevalence increased across age groups for both men and women although the percentage increase

**Table 1. Sex-disaggregated study population demographics, perceptions, and behaviours.**

| Variable | Overall, N=761 (%) | Female, N=481 (%) | Male, N=280 (%) | *p*-value* |
|---|---|---|---|---|
| **Demographic** | | | | |
| Age | | | | 0.6 |
|   18-30 | 202 (26.5) | 124 (25.8) | 78 (27.9) | |
|   31-45 | 270 (35.5) | 169 (35.1) | 101 (36.1) | |
|   46-60 | 191 (25.1) | 120 (24.9) | 71 (25.4) | |
|   >60 | 98 (12.9) | 68 (14.1) | 30 (10.7) | |
| Race | | | | 0.7 |
|   Black | 386 (50.71) | 241 (50.1) | 145 (51.8) | |
|   Other | 48 (6.3) | 33 (6.9) | 15 (5.4) | |
|   Pardo | 327 (43.0) | 207 (43.0) | 120 (42.9) | |
| Highest level of school studied | | | | >0.9 |
|   Secondary | 161 (21.2) | 102 (21.2) | 59 (21.1) | |
|   Primary | 600 (78.8) | 379 (78.8) | 221 (78.9) | |
| Employment | | | | <0.001 |
|   Unemployed | 370 (48.7) | 276 (57.5) | 94 (33.7) | |
|   Formal | 159 (20.9) | 65 (13.5) | 94 (33.7) | |
|   Informal | 230 (30.3) | 139 (29.0) | 91 (32.6) | |
|   Unknown | 2 | 1 | 1 | |
| Occupation type (among employed) | | | | 0.8 |
|   High-risk | 95 (24.4) | 49 (24.0) | 46 (24.9) | |
|   Other | 294 (75.6) | 155 (76.0) | 139 (75.1) | |
| **Serostatus** | | | | |
|   Positive | 86 (11.3) | 45 (9.4) | 41 (14.6) | 0.03 |
|   Negative | | | | |
| **Perceptions** | | | | |
| Perceived severity of leptospirosis | | | | 0.8 |
|   Extremely serious | 649 (87.0) | 411 (86.7) | 238 (87.5) | |
|   Less serious | 97 (13.0) | 63 (13.3) | 34 (12.5) | |
|   Unknown | 15 | 7 | 8 | |
| **Behaviours (conducted in last 6 months)** | | | | |
| Walked through floodwater | | | | 0.2 |
|   Yes | 186 (24.9) | 126 (26.6) | 60 (22.1) | |
|   No | 560 (75.1) | 348 (73.4) | 212 (77.9) | |
|   Unknown | 15 | 7 | 8 | |
| Walked through sewage water | | | | 0.7 |
|   Yes | 161 (21.6) | 100 (21.1) | 61 (22.3) | |
|   No | 586 (78.4) | 374 (78.9) | 212 (77.7) | |
|   Unknown | 14 | 7 | 7 | |
| Could wear boots during flooding | | | | <0.001 |
|   Yes | 222 (29.7) | 86 (18.1) | 136 (49.8) | |
|   No | 526 (70.3) | 389 (81.9) | 137 (50.2) | |
|   Unknown | | | | |
| Walked barefoot outside of home | | | | 0.7 |
|   Yes | 174 (23.3) | 108 (22.8) | 66 (24.2) | |
|   No | 573 (76.7) | 366 (77.2) | 207 (75.8) | |

*(Continued)*

| Variable | Overall, N = 761¹ (%) | Female, N = 481¹ (%) | Male, N = 280¹ (%) | *p*-value* |
|---|---|---|---|---|
| Unknown | 14 | 7 | 7 | |
| Walked through mud | | | | 0.4 |
| Yes | 197 (26.3) | 120 (25.3) | 77 (28.2) | |
| No | 551 (73.7) | 355 (74.7) | 196 (71.8) | |
| Unknown | 13 | 6 | 7 | |

*Pearson's Chi-squared.

was greater in men (30% increase across 18–30 to >60 age groups in men, vs. 8% increase across 18–30 and >60 age groups in women). Seroprevalence was also lower among both men and women who perceived leptospirosis as an extremely serious disease than those who didn't (13% vs. 27% in men and 9% vs. 14% in women). The sex-disaggregated descriptive analysis of seroprevalence across all risk factors is included in S1 Table.

**Regression analyses.** Uni- and multivariable analyses were used to estimate the total causal effect of each exposure on seropositivity. The univariable analysis is included in S2 Table. In the multivariable analysis, men had 1.76 (95% CI: 1.11, 2.79) times the odds of *Leptospira* seropositivity than women (Table 2).

Older age was associated with increased odds of seropositivity in the male-restricted and combined models, but not in the female-restricted model. In the male-restricted model, each year of age in men was associated with 1.04 (95% CI: 1.01, 1.06) times the odds of seropositivity. In the combined model, each year of age was associated with 1.03 (95% CI: 1.01, 1.04) times the odds of seropositivity. A similar estimate was found in the female-restricted model, as each year of age was associated with 1.02 (95% CI: 0.99, 1.05) times the odds of seropositivity, however the association was not significant at the 5% level. A test for interaction showed no evidence that the association of age with seropositivity differed significantly between genders (*p*-value of interaction = 0.2).

Greater perceived severity was associated with reduced odds of seropositivity in the male-restricted and combined models, but not in the female-restricted model. In the male-restricted model, men who perceived leptospirosis as extremely serious had 0.38 (95% CI: 0.15, 0.99) times the odds of being seropositive than men who perceived leptospirosis as less serious. In the combined model, those that perceived leptospirosis as extremely serious had 0.55 (95% CI: 0.30, 0.99) times the odds of being seropositive than those that perceived leptospirosis as less serious. A similar estimate was found in the female-restricted model, with women that perceived leptospirosis as extremely serious having 0.54 (95% CI: 0.18, 1.60) times the odds of being seropositive than women who perceived leptospirosis as less serious, however the association was not significant at the 5% level. The associations in the male- and female-restricted models are visualised in Fig 6A. A test for interaction showed no evidence that the association of perceived severity with seropositivity differed significantly between genders (*p*-value of interaction = 0.3).

Among employed participants, high-risk occupations were associated with increased odds of seropositivity in the male-restricted model, but not in the combined or female-restricted models. In the male-restricted model, men who had high-risk occupations had 3.92 (95% CI: 1.17, 13.2) times the odds of being seropositive than those that had other occupations. In the combined model, those in high-risk occupations had 1.60 (95% CI: 0.72, 3.54) times the odds of being seropositive than those in other occupations. A conflicting estimate was found in the female-restricted model, as women in high-risk occupations had 0.92 (95% CI: 0.23, 2.92) times the odds of being seropositive than those in other occupations, although the association was not significant at the 5% level. Despite the difference in point estimates, a test for interaction presented no evidence that the association of high-risk occupations with seropositivity differed significantly between genders (*p*-value of interaction = 0.07).

 

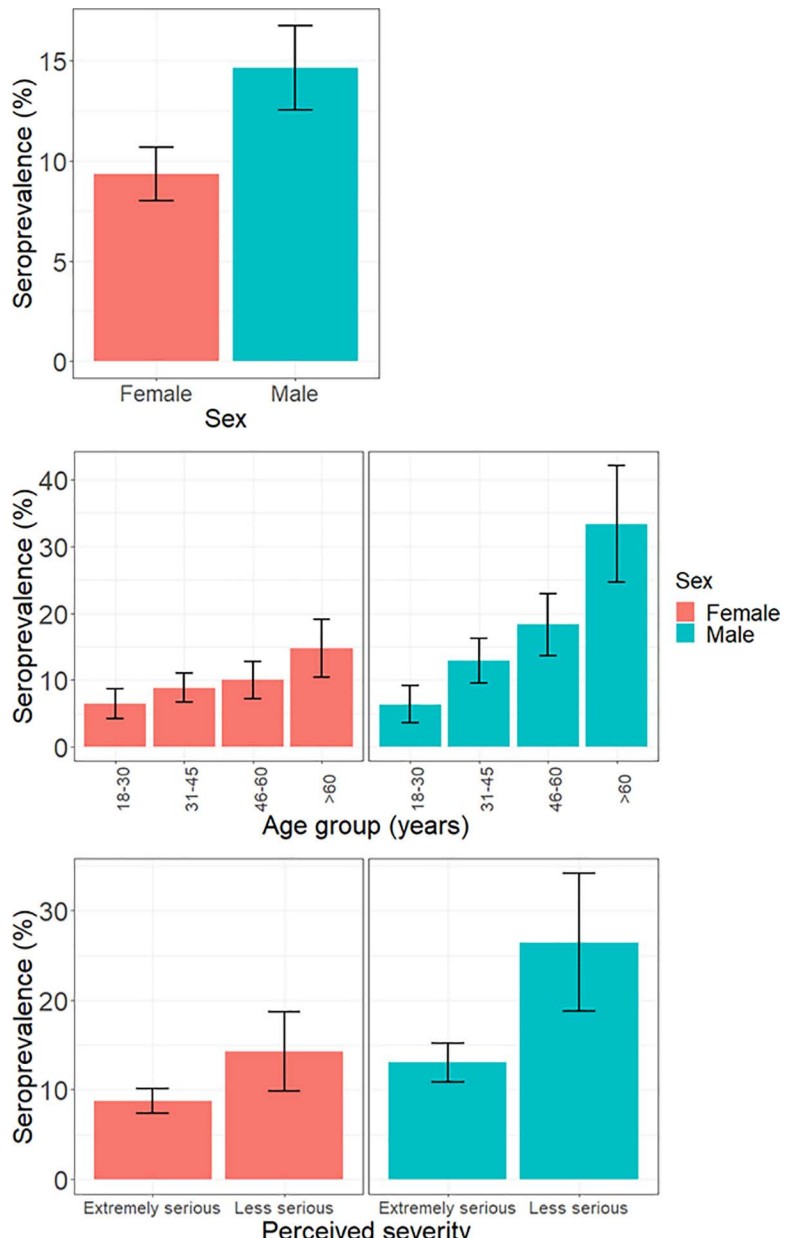

**Fig 5. Sex-disaggregated seroprevalence across sex, age group, and perceived severity.**

Other exposures, such as race, education, employment, and behaviours, were not associated with seropositivity in the combined or sex-disaggregated models.

### Analysis 2: Perceived severity as a determinant of high-risk behaviours

To further investigate the finding that perceived severity of leptospirosis was associated with seropositivity and explore its role as a driver of seropositivity through behaviours (as illustrated in Fig 3), we examined the association between perceived severity and high-risk behaviours.

**Table 2. Total causal effect estimates for the effect of each exposure on seropositivity, shown for the combined and sex-disaggregated multivariable logistic regression models.**

| Exposure | Combined | | | Sex-disaggregated | | | | | | |
|---|---|---|---|---|---|---|---|---|---|---|
| | | | | **Female-restricted** | | | **Male-restricted** | | | **p-value of interaction** |
| | n | aOR (95% CI) | p-value | n | aOR (95% CI) | p-value | n | aOR (95% CI) | p-value | |
| **Sociodemographic** | | | | | | | | | | |
| Gender* | 761 | | | | | | | | | |
| Female | | REF | | | N/A | | | N/A | | |
| Male | | 1.76 (1.11, 2.79) | 0.015 | | | | | | | N/A |
| Age (increase per year)** | 761 | 1.03 (1.01, 1.04) | <0.001 | 481 | 1.02 (0.99, 1.05) | 0.12 | 280 | 1.04 (1.01, 1.06) | 0.002 | 0.2 |
| Race*** | 761 | | | 481 | | | 280 | | | |
| Black | | REF | | | REF | | | REF | | |
| Other | | 0.86 (0.32, 2.30) | 0.8 | | 1.33 (0.31, 5.81) | 0.7 | | 0.43 (0.05, 3.58) | 0.4 | 0.4 |
| Pardo | | 0.90 (0.56, 1.44) | 0.7 | | 0.99 (0.43, 2.24) | >0.9 | | 0.76 (0.38, 1.54) | 0.5 | 0.6 |
| Highest level of school studied **** | 761 | | | 481 | | | 280 | | | |
| Secondary school | | REF | | | REF | | | REF | | |
| Primary school | | 1.29 (0.66, 2.53) | 0.5 | | 1.68 (0.55, 5.13) | 0.4 | | 1.02 (0.38, 2.73) | >0.9 | 0.5 |
| Employment status (in the last week)*^ | 759 | | | 480 | | | 279 | | | |
| Formally employed | | REF | | | REF | | | REF | | |
| Unemployed | | 1.21 (0.64, 2.30) | 0.6 | | 0.93 (0.31, 2.79) | >0.9 | | 1.55 (0.63, 3.82) | 0.3 | 0.3 |
| Informally employed | | 0.91 (0.46, 1.82) | 0.8 | | 0.93 (0.28, 2.79) | >0.9 | | 0.80 (0.31, 2.08) | 0.7 | 0.9 |
| Occupation (among employed)*^^ | 382 | | | 200 | | | 182 | | | |
| Other | | REF | | | REF | | | REF | | |
| High-risk | | 1.60 (0.72, 3.54) | 0.3 | | 0.82 (0.23, 2.92) | 0.8 | | 3.92 (1.17, 13.2) | 0.027 | 0.068 |
| **Perceptions of leptospirosis** | | | | | | | | | | |
| Perceived severity of leptospirosis*^^^ | 725 | | | 462 | | | 261 | | | |
| Less serious | | REF | | | REF | | | REF | | |
| Extremely serious | | 0.55 (0.30, 0.99) | 0.047 | | 0.54 (0.18, 1.60) | 0.3 | | 0.38 (0.15, 0.99) | 0.047 | 0.3 |
| **Behaviours (conducted in the last 6 months)** | | | | | | | | | | |
| Walked through floodwater$ | 708 | | | 452 | | | 252 | | | |
| Rarely or never | | REF | | | REF | | | REF | | |
| Frequently | | 0.95 (0.44, 2.09) | >0.9 | | 1.03 (0.37, 2.90) | >0.9 | | 0.75 (0.21, 2.75) | 0.7 | 0.8 |
| Walked through sewage water$* | 716 | | | 452 | | | 255 | | | |
| Rarely or never | | REF | | | REF | | | REF | | |
| Frequently | | 0.90 (0.40, 2.01) | 0.8 | | 0.64 (0.18, 2.28) | 0.5 | | 1.17 (0.34, 4.01) | 0.8 | 0.7 |
| Could wear boots during flooding$** | 716 | | | 464 | | | 264 | | | |
| Yes | | REF | | | REF | | | REF | | |
| No | | 1.03 (0.61, 1.74) | >0.9 | | 1.01 (0.36, 2.83) | >0.9 | | 1.01 (0.50, 2.06) | >0.9 | >0.9 |
| Walked barefoot outside of home$*** | 729 | | | 452 | | | 255 | | | |
| Rarely or never | | REF | | | REF | | | REF | | |

*(Continued)*

**Table 2.** (Continued)

| Exposure | Combined | | | Sex-disaggregated | | | | | | p-value of interaction |
|---|---|---|---|---|---|---|---|---|---|---|
| | | | | Female-restricted | | | Male-restricted | | | |
| | n | aOR (95% CI) | p-value | n | aOR (95% CI) | p-value | n | aOR (95% CI) | p-value | |
| Frequently | | 1.62 (0.93, 2.84) | 0.086 | | 1.67 (0.71, 3.91) | 0.2 | | 1.71 (0.72, 4.03) | 0.2 | 0.9 |
| Walked through mud$**** | 727 | | | 463 | | | 258 | | | |
| Rarely or never | | REF | | | REF | | | REF | | |
| Frequently | | 1.29 (0.76, 2.18) | 0.4 | | 1.42 (0.69, 2.95) | 0.3 | | 1.25 (0.55, 2.82) | 0.6 | 0.7 |

REF: Reference group. Adjustments: *neighbourhood, race, age; **neighbourhood, race, gender; ***age, gender; ****neighbourhood, race, age, gender; *^neighbourhood, race, age, gender, marital status, education; *^^neighbourhood, race, age, gender, education, food insecurity, employment status; *^^^neighbourhood, race, age, gender, socioeconomic status; $neighbourhood, race, age, gender, socioeconomic status, marital status, household environments, ownership of boots, walked through sewage, walked barefoot; $*neighbourhood, race, age, gender, marital status, socioeconomic status, household environments, ownership of boots, walked barefoot, walked through food water; $**race, age, education, gender, employment, food insecurity; $***neighbourhood, race, age, gender, marital status, socioeconomic status, household environments, ownership of boots, walked through flood water, walked through sewage; &****neighbourhood, race, age, gender, socioeconomic status, ownership of boots.

**Descriptive.** The sex-disaggregated descriptive analysis of the distribution of high-risk behaviours across perceived severity is included in S3 Table. Notably, a smaller proportion of men who perceived leptospirosis as extremely serious reported walking barefoot outside of the home than those who perceived leptospirosis as less serious (21% vs. 41%).

**Regression analyses.** Uni- and multivariable analyses were used to estimate the total causal effect of perceived severity on the risk of performing high-risk behaviours. The univariable analysis is included in S4 Table. The full multivariable regression analysis exploring the association of perceived severity and behaviours is shown in S5 Table, with two key findings shown in Fig 6B.

Perceived severity was associated with lower odds of walking barefoot outside of the home in the male-restricted model but not in the female-restricted model (Fig 6B), or the combined model. In the male-restricted models, men who perceived leptospirosis as extremely serious had 0.24 (95% CI: 0.08, 0.76) times the odds of walking barefoot outside of the home compared with men who perceived leptospirosis as less serious. In the female-restricted models, women who perceived leptospirosis as extremely serious had 1.15 (95% CI: 0.53, 2.50) times the odds of walking barefoot than women who perceived leptospirosis as less serious, however the association was not significant at the 5% level. A test for interaction provided further evidence to support that the association between perceived severity and walking barefoot differed significantly between genders (p-value of interaction = 0.007).

Perceived severity was also associated with reduced odds of walking through sewage water in the male-restricted model but not in the female restricted model (Fig 6B), or the combined model. In the male-restricted models, men who perceived leptospirosis as extremely serious had 0.41 (95% CI: 0.17, 1.00) times the odds of walking through sewage water than men who perceived leptospirosis as less serious. In the female-restricted models, women who perceived leptospirosis as extremely serious had 0.80 (95% CI: 0.41, 1.62) times the odds of walking through sewage water than women who perceived leptospirosis as less serious, however the association was not significant at the 5% level. A test for interaction showed no evidence that the association of perceived severity with walking through sewage differed by gender (p-value of interaction = 0.3).

**Sub-analysis: Perceived severity and high-risk behaviours across age groups.** Further stratification by age revealed distinct patterns of association between perceived severity of leptospirosis and walking barefoot or walking through sewage water, particularly in males (S4 Fig). In the male-restricted model, men aged 18–30 and 31–45 years who perceived leptospirosis as extremely serious had 0.09 (95% CI: 0.01, 0.62) and 0.10 (95% CI: 0.01, 0.53) times the odds of walking barefoot respectively, compared to men in the same age groups who perceived leptospirosis as less serious. In contrast, no evidence of an association was observed in men aged 46–60 years (OR: 1.13; 95% CI: 0.11, 28.56).

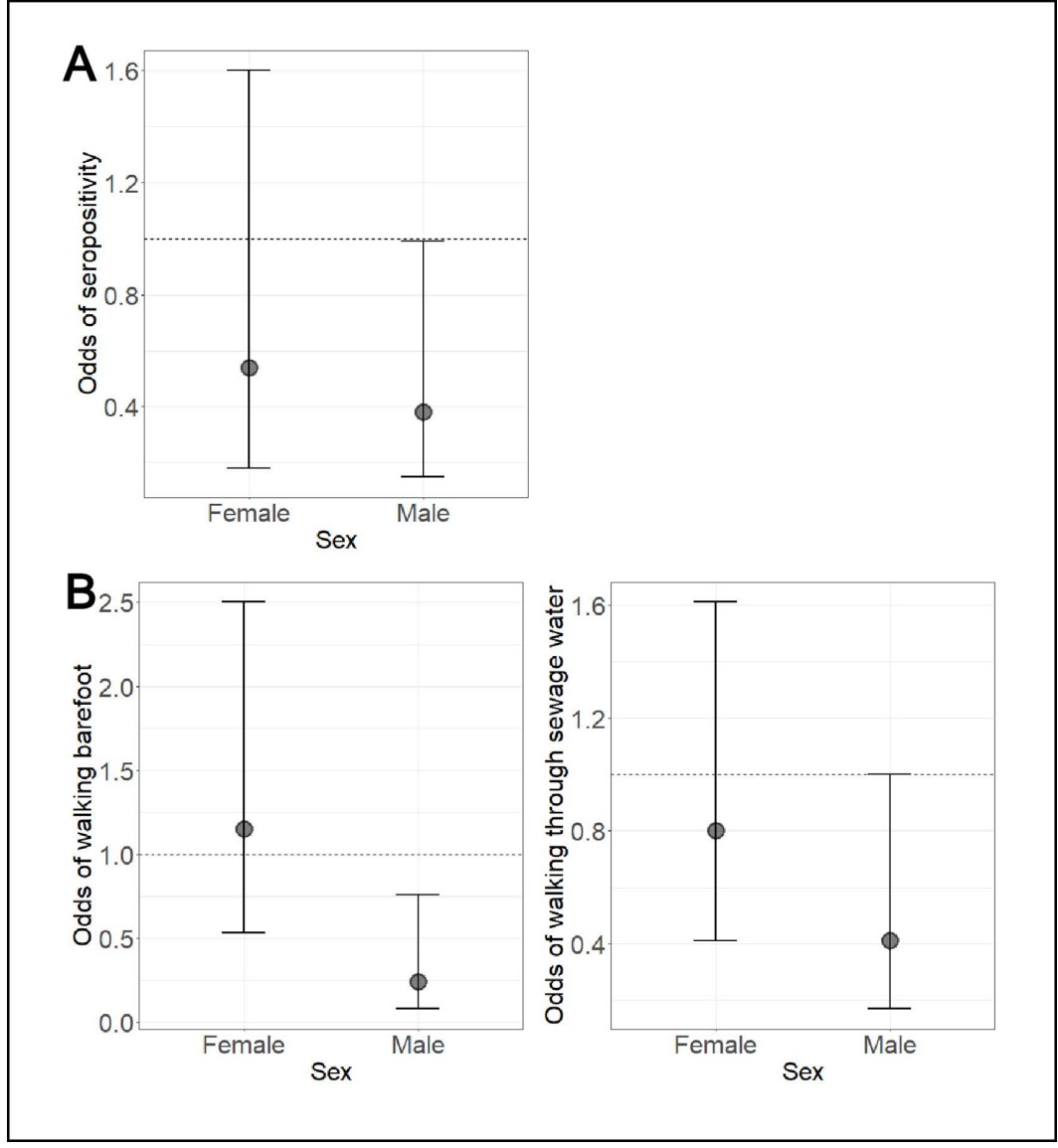

**Fig 6. Total causal effects of perceived severity with A) seropositivity and B) high-risk behaviours.** Odds ratios are shown for females or males who perceived leptospirosis as extremely serious compared to those of the same gender who perceived leptospirosis as less serious.

Additionally, men aged 31–45 years who perceived leptospirosis as extremely serious had 0.15 (95% CI: 0.03, 0.79) times the odds of walking through sewage water compared to men who perceived leptospirosis as less serious, however there was no evidence of an association in men aged 18–30 or 45–60 years. There was no observed difference in associations between perceived severity and high-risk behaviours by age group across any female-restricted model.

## Discussion

We present the first sex-disaggregated study to investigate gender-specific infection risk of *Leptospira*. We examined sociodemographic, perception, and behaviour risk factors to explain seroprevalences of 14.6% and 9.4% in men and women, respectively. High proportions of both genders perceived leptospirosis as an extremely serious disease, but only

men who viewed it as extremely serious had lower odds of being seropositive than men who perceived it as less serious. While a similar proportion of men and women had high-risk occupations among employed participants, only men in high-risk occupations had higher odds of being seropositive than men in other occupations. Men who perceived leptospirosis as extremely serious had lower odds of walking barefoot or through sewage water, an association not found in women, with evidence of gender modifying the relationship between perceived severity and walking barefoot. Taken together, our findings identified distinct gender-based associations not visible in combined analyses, indicating that high-risk occupations, perceptions, and behaviours drive differential infection risk by gender. These results demonstrate the benefit of sex-disaggregated frameworks in disentangling complex, gender-determined infection risks.

The finding that men who perceived leptospirosis as extremely serious were less likely to walk through sewage or walk barefoot outside of the home than those who did not, suggests that perceptions of disease severity may influence disease-specific preventive behaviours. This aligns with the Rational Model Theory [51]. Walking through sewage water and walking barefoot are commonly cited risk factors for leptospirosis [4,52], particularly the latter with 11 out of 12 studies included in one systematic review identifying walking barefoot outside as a risk factor [16]. These behaviours are particularly high-risk in these study areas as the communities are built over open sewers that often overflow with heavy rain. This can disperse leptospires into the surrounding soil or water, increasing the risk of human infection through broken skin or cuts on barefeet.

Walking barefoot and walking through sewage water were the only behaviours associated with perceived severity, which could be explained by them being easier to moderate than the other measured behaviours in this analysis, such as contact with flood water and mud. For residents of flood-prone marginalised communities with unpaved streets, these sources of environmental risk are often very challenging to avoid when moving within the community, and exposure to them may be driven more strongly by household location than perceptions and attitudes to risk. Furthermore, the ability to wear boots during flooding is likely driven by socioeconomic circumstances, as was also found in the previous KAP study of leptospirosis risks [8]. Conversely, participants may be able to avoid areas where sewers are known to overflow and avoid walking barefoot,and therefore these behaviours are more directly within a person's control.

We found that sex was an effect modifier for the relationship between perceived severity and walking barefoot, with no evidence in the female-restricted model of a statistical association. We hypothesised that the lack of association between perceived severity and walking barefoot in the female-restricted model could be explained by women having much more limited access to boots than men (18% vs. 50%). For this reason, we tested to see whether the gender interaction could be explained by this difference in boot ownership but found no evidence of an interaction of boot ownership with perceived severity ($p$-value of interaction = 0.3). While the determinants underlying this gender interaction effect remain uncertain, this finding highlights the importance of conducting sex-disaggregated or gender-interaction analyses to identify gender-specific relationships that are obscured in combined models.

We observed that the relationship between perceived disease severity and high-risk behaviours varied across age groups in males. Men aged 31–45 years who perceived leptospirosis as extremely serious had lower odds of engaging in risky behaviours, such as walking barefoot or through sewage water. Similarly, men aged 18–30 years who perceived leptospirosis as extremely serious also showed lower odds of walking barefoot. These findings suggest that health interventions should be tailored to address age-specific perceptions and behaviours, particularly for men in these age groups, who may benefit from messages promoting the risk of leptospirosis.

Despite perceived severity and seropositivity being associated in men and the point estimate for women also being in the same direction, none of the high-risk behaviours analysed were associated with *Leptospira* seropositivity. This may be indicative of the relative contributions of two infection risks: 1) periods of intense flooding exposure in and around the household, and 2) low-intensity exposure to soil and mud. It is likely that our study population had a homogeneous and high environmental risk from flooding events because the study areas were relatively small and were selected to include households located in the lowest sections of the communities, where the risk of leptospirosis is known to be highest [15,30]. Therefore, environmental exposure due to heavy flooding may ultimately mean that the effect of preventive behaviours,

such as wearing shoes, is limited. To better understand this causal pathway, future studies should consider studying areas with greater variation in environmental risk to identify behaviours with a substantial impact on individual risk.

The absence of statistical associations observed in female-restricted models suggests that exposures in women may not be fully captured. Given that male gender is commonly cited as a risk factor in leptospirosis studies in Salvador and globally, the measured exposures may be biased toward male gender norms [4,12,13]. To address this, potential exposures aligned with female cultural norms in Salvador should be incorporated to distinguish infection risks in women. For instance, women in Salvador are traditionally associated with limited mobility in the environment and engage in domestic roles within the home [22]. Consequently, exposure to *Leptospira* may occur in the household environment during these activities. Future serosurveys should thus consider expanding the scope of questions to encompass behavioural exposures that may be more prevalent among women.

Due to the neglected status of leptospirosis and the broader gap in sex-disaggregated CSID research, limited gender-specific data hinders our understanding of infection risks and behavioural exposures in women [10,21,53,54]. To address this gap, qualitative focus group discussions segregated by gender may help to shape hypotheses surrounding exposures in women, which could subsequently be measured and tested in serosurveys [55]. This mixed-methods approach allows for open-ended questioning to capture unanticipated information and has been of benefit in South Africa to reveal gender-based exposures to leptospirosis [56,57]. Identifying and measuring exposures more prevalent in women could help tailor interventions to both men and women, thus improving our understanding and control of leptospirosis transmission.

### Limitations

A limitation of our study is the use of serostatus as an indicator of recent infection, which is complicated by the persistence of antibodies that can remain detectable for several years after exposure [58]. Since our exposure-risk behaviours were self-reported over the past six months, this persistence makes it difficult to interpret seropositivity as a direct reflection of recent behaviours. However, the responses collected regarding the previous 6 months may be indicative of behaviours and perceptions over longer periods, as these are unlikely to change significantly. While it is conventionally characterised that younger adult age groups are at higher risk of *Leptospira* infection [12,13], we observed a linear increase in risk with age, which is consistent with previous studies in Salvador that identified the same trend [4,15]. The role of antibody persistence in this age-related pattern further emphasises the challenges of using serostatus as a sole measure of recent infection risk. Moreover, serostatus may not capture gender inequalities in disease burden, particularly as there is evidence that progression of leptospirosis to SPHS is more common in women than men [20]. Therefore, inclusion of broader disease burden metrics that encompass both prevalence and outcomes, may better address gender disparities in leptospirosis research.

The study also has limitations related to the sample size and self-reporting of behaviours. Constraints on sample size, particularly the smaller number of seropositive women, may affect the statistical power of our findings, potentially limiting the ability to detect certain gender-based associations. The fact that the point estimate for women who perceived leptospirosis as extremely serious was greater than 1 for the odds of walking barefoot, but with much greater uncertainty (OR: 1.23; 95% CI: 0.57, 2.67), illustrates this point. Additionally, self-reporting of behaviours could introduce recall or social desirability bias during interviews, impacting the accuracy of data and thereby influencing the associations observed in our analyses. Future studies with larger, more representative samples and alternative data collection methods may improve both generalisability and accuracy.

Finally, while our study provides insights into gender-related exposure pathways, the use of binary gender categories of men and women presents a limitation. Future studies should consider broader data collection methods that accommodate diverse gender identities and explore the intersectional effects of other social determinants of health, such as race and socioeconomic status.

## Conclusion

As the first sex-disaggregated study on *Leptospira* infection, our findings highlight differences in exposure risks between men and women, driven by perceptions of disease severity and high-risk behaviours that shape gender-specific vulnerabilities to leptospirosis. We found that occupational exposure plays a role in seropositivity among men, and that high-risk behaviours, influenced by disease perceptions, increase their exposure to contaminated environments. In contrast, while women shared some of these exposures, the lack of statistical associations in female-restricted models suggests that additional, gender-specific factors influencing infection risk in women remain unidentified. Further research is needed to address this gap, particularly to understand how women's behaviours and perceptions of leptospirosis contribute to their infection risk.

To effectively reduce the risk of *Leptospira* infection in Salvador, public health services must implement targeted interventions that address both immediate and long-term drivers of infection. For men, prioritising occupational health and safety measures may be important. Additionally, health promotion programs should aim to raise awareness about the severity of leptospirosis through effective communication strategies, which can help reduce high-risk behaviours such as walking barefoot and exposure to floodwaters. While these measures may also benefit women, further research is essential to identify specific exposure pathways for women and develop tailored public health interventions. Community dissemination of these findings is important, and organisations like the Centre for the Control of Zoonoses (CCZ) could play a key role in exploring the influence of perceptions as their health agents provide door-to-door health advice.

Our findings underscore the value of sex-disaggregated frameworks in identifying gender-specific risks. Integrating broader intersectional factors, such as biological, social, and environmental drivers, can enhance the understanding of *Leptospira* infection risk and other CSIDs. This approach will support the development of equitable, climate-sensitive interventions that address the differential risks experienced by men, women, boys, girls, and other gender identities.

## Supporting information

**S1 Checklist. PLOS inclusivity in global research checklist.**
(DOCX)

**S1 Fig. Full version of the DAG used in analysis, also available at** https://dagitty.net/dags.html?id=XxPTXytr#**.** DWA: Domestic waste accumulation. < 10m from DWA and open sewer refer to household distance from risk factors.
(DOCX)

**S2 Fig. GAMs of age with seroprevalence response variable.** GAMs were built using univariable models. Shaded area corresponds to 95% CI.
(DOCX)

**S3 Fig. GAMs of age with high-risk behaviour response variables.** Position of knots used to model non-linear relationships. GAMs were built using univariable models. Shaded area corresponds to 95% CI.
(DOCX)

**S4 Fig. Total causal effects of perceived severity with A) walking barefoot outside the home and B) walking through sewage water, stratified by age.** Odds ratios are presented for females and males who perceived leptospirosis as extremely serious, compared to those of the same gender and age group who perceived leptospirosis as less serious. The plots are displayed on a log scale due to the high uncertainty in some of the estimates.
(DOCX)

**S1 Table. Sex-disaggregated descriptive analysis of seroprevalence across risk factors.**
(DOCX)

**S2 Table. Sex-disaggregated univariable logistic regression analysis of seroprevalence across risk factors.**
(DOCX)

**S3 Table. Descriptive analysis of prevalence of behaviours across perceived severity.**
(DOCX)

**S4 Table. Sex-disaggregated univariable logistic regression analysis of the association of perceived severity with high-risk behaviours.**
(DOCX)

**S5 Table. Total causal effect estimates for the effect of perceived severity on the risk of performing high-risk behaviours, shown for the combined and sex-disaggregated multivariable logistic regression models.**
(DOCX)

**S6 Table. Descriptive analysis of age across explanatory variables in men (A) and women (B).**
(DOCX)

## Acknowledgments

We thank the residents and community leaders of Nova Sussuarana, Arenoso, Calabetão, and Jardim Santo Inácio/Mata Escura, for their support and participation in this study. We would also like to thank the community driving group (GIC-Grupo Impulsor Comunitario) in the communities Nova Sussuarana and Arenoso for their support.

## Author contributions

**Conceptualization:** Ellie A. Delight, Diogo César de Carvalho Santiago, Fabiana Almerinda G. Palma, Yeimi Alzate López, Akanksha A. Marphatia, Cleber Cremonese, Federico Costa, Max T. Eyre.

**Data curation:** Daiana de Oliveira, Fabio Neves Souza, Cleber Cremonese.

**Formal analysis:** Ellie A. Delight.

**Funding acquisition:** Albert I. Ko, Federico Costa.

**Investigation:** Ellie A. Delight, Diogo César de Carvalho Santiago, Fabiana Almerinda G. Palma, Cleber Cremonese.

**Methodology:** Ellie A. Delight, Diogo César de Carvalho Santiago, Fabiana Almerinda G. Palma, Daiana de Oliveira, Yeimi Alzate López, Federico Costa, Max T. Eyre.

**Project administration:** Max T. Eyre.

**Resources:** Diogo César de Carvalho Santiago, Daiana de Oliveira, Juliet Oliveira Santana, Mitermayer G. Reis, Cleber Cremonese, Max T. Eyre.

**Supervision:** Arata Hidano, Cleber Cremonese, Federico Costa, Max T. Eyre.

**Visualization:** Ellie A. Delight, Juliet Oliveira Santana.

**Writing – original draft:** Ellie A. Delight, Max T. Eyre.

**Writing – review & editing:** Ellie A. Delight, Diogo César de Carvalho Santiago, Fabiana Almerinda G. Palma, Daiana de Oliveira, Fabio Neves Souza, Juliet Oliveira Santana, Arata Hidano, Yeimi Alzate López, Mitermayer G. Reis, Albert I. Ko, Akanksha A. Marphatia, Cleber Cremonese, Federico Costa, Max T. Eyre.

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
