## [Decision Letter · Decision Letter 0]

PGPH-D-24-01780

Gender differences in the perception of leptospirosis severity, behaviours, and *Leptospira*  exposure risk in urban Brazil: a cross-sectional study

Dear Dr. Costa,

Thank you for submitting your manuscript to PLOS Global Public Health. After careful consideration, we feel that it has merit but does not fully meet PLOS Global Public Health’s publication criteria as it currently stands. Therefore, we invite you to submit a revised version of the manuscript that addresses the points raised during the review process.

One of the aspects that merit further attention refers to the need to share the actual dataset and the scripts of the analyses as one of the reviewers indicated that he could not find the expected results. Please do so as appropriate.

We look forward to receiving your revised manuscript.

Kind regards,

André Machado Siqueira, M.D., MSc, Ph.D

Academic Editor

Journal Requirements:

 1. Please include a complete copy of PLOS’ questionnaire on inclusivity in global research in your revised manuscript. Our policy for research in this area aims to improve transparency in the reporting of research performed outside of researchers’ own country or community. The policy applies to researchers who have travelled to a different country to conduct research, research with Indigenous populations or their lands, and research on cultural artefacts. The questionnaire can also be requested at the journal’s discretion for any other submissions, even if these conditions are not met.  Please find more information on the policy and a link to download a blank copy of the questionnaire here: https://journals.plos.org/globalpublichealth/s/best-practices-in-research-reporting. Please upload a completed version of your questionnaire as Supporting Information when you resubmit your manuscript. 2. Please amend your detailed Financial Disclosure statement. This is published with the article. It must therefore be completed in full sentences and contain the exact wording you wish to be published. **Please only choose the relevant sentences from below** 1. Please clarify all sources of funding (financial or material support) for your study. List the grants (with grant number) or organizations (with url) that supported your study, including funding received from your institution. 2. State the initials, alongside each funding source, of each author to receive each grant.3. State what role the funders took in the study. If the funders had no role in your study, please state: “The funders had no role in study design, data collection and analysis, decision to publish, or preparation of the manuscript.”4. If any authors received a salary from any of your funders, please state which authors and which funders. If you did not receive any funding for this study, please simply state: “The authors received no specific funding for this work.” 3. Please send a completed 'Competing Interests' statement, including any COIs declared by your co-authors. If you have no competing interests to declare, please state "The authors have declared that no competing interests exist". Otherwise please declare all competing interests beginning with the statement "I have read the journal's policy and the authors of this manuscript have the following competing interests:" 4. "Figure 3": Please confirm whether you drew the images / clip-art within the figure panels by hand. If you did not draw the images, please provide (a) a link to the source of the images or icons and their license / terms of use; or (b) written permission from the copyright holder to publish the images or icons under our CC-BY 4.0 license. Alternatively, you may replace the images with open source alternatives. See these open source resources you may use to replace images / clip-art:- https://commons.wikimedia.org-
https://openclipart.org/ 5. "Figure 1": please (a) provide a direct link to the base layer of the map (i.e., the country or region border shape) and ensure this is also included in the figure legend; and (b) provide a link to the terms of use / license information for the base layer image or shapefile. We cannot publish proprietary or copyrighted maps (e.g. Google Maps, Mapquest) and the terms of use for your map base layer must be compatible with our CC-BY 4.0 license.  Note: if you created the map in a software program like R or ArcGIS, please locate and indicate the source of the basemap shapefile onto which data has been plotted. If your map was obtained from a copyrighted source please amend the figure so that the base map used is from an openly available source. Alternatively, please provide explicit written permission from the copyright holder granting you the right to publish the material under our CC-BY 4.0 license. Please note that the following CC BY licenses are compatible with PLOS license: CC BY 4.0, CC BY 2.0 and CC BY 3.0, meanwhile such licenses as CC BY-ND 3.0 and others are not compatible due to additional restrictions.  If you are unsure whether you can use a map or not, please do reach out and we will be able to help you. The following websites are good examples of where you can source open access or public domain maps: * U.S. Geological Survey (USGS) - All maps are in the public domain. (http://www.usgs.gov) * PlaniGlobe - All maps are published under a Creative Commons license so please cite “PlaniGlobe, http://www.planiglobe.com, CC BY 2.0” in the image credit after the caption. (http://www.planiglobe.com/?lang=enl) * Natural Earth - All maps are public domain. (http://www.naturalearthdata.com/about/terms-of-use/)

Additional Editor Comments (if provided):

Reviewers' comments:

Reviewer's Responses to Questions

**Comments to the Author**

1. Does this manuscript meet PLOS Global Public Health’s publication criteria?

Reviewer #1: Yes

Reviewer #2: No

2. Has the statistical analysis been performed appropriately and rigorously?

Reviewer #1: I don't know

Reviewer #2: I don't know

3. Have the authors made all data underlying the findings in their manuscript fully available (please refer to the Data Availability Statement at the start of the manuscript PDF file)?

Reviewer #1: Yes

Reviewer #2: Yes

4. Is the manuscript presented in an intelligible fashion and written in standard English?

Reviewer #1: Yes

Reviewer #2: Yes

Reviewer #1: From the hypothesis to the final results, the study "Gender differences in the perception of leptospirosis severity, behaviours, and Leptospira exposure risk in urban Brazil" shows how gender affects (cultural) perception and risk behavior among humans and the effects of this on the exposure risk and severity of diseases. This is significant for health education, and generally preventive medicine/care.

The study design is a cross sectional study which involved some parts of Brazil. However, the researchers failed to exclusively state how they came about the sample size of 761.

Also, with regards to the in-text citation: for Vancouver style, the reference number is placed to the left of or inside of colons and semi-colons; but placed to the right of or outside of full stop.

Reviewer #2: This is an interesting article, though its scope appears to be more locally focused. I believe it would benefit from including recommendations that bridge academia with health services, public health surveillance, and strategies for community empowerment and participation in health.

Several points could benefit from further elaboration. While the methodology is comprehensive, it could be made more concise. Additionally, I felt that more information on the sampling aspects of the study would enhance the clarity of the methodology.

Further detailed comments can be found in the attached document.

**Do you want your identity to be public for this peer review?** For information about this choice, including consent withdrawal, please see our Privacy Policy

Reviewer #1: **Yes: ** Abdulmalik Opeyemi Adeyemo

Reviewer #2: No

---

## [Decision Letter · Decision Letter 1]

Gender differences in *Leptospira*  exposure risk, perceptions of disease severity, and high-risk behaviours in Salvador, Brazil: a cross-sectional study (2021-2022)

PGPH-D-24-01780R1

Dear Frederico Costa

We are pleased to inform you that your manuscript 'Gender differences in *Leptospira*  exposure risk, perceptions of disease severity, and high-risk behaviours in Salvador, Brazil: a cross-sectional study (2021-2022)' has been provisionally accepted for publication in PLOS Global Public Health.

Thank you again for supporting Open Access publishing; we are looking forward to publishing your work in PLOS Global Public Health. It has been unfortunately a very hard last few months for many of our colleagues, who as a result wrote that they could not review this paper, delaying the process.

Best regards,

Megan Coffee, MD, PhD

Academic Editor

Reviewer Comments (if any, and for reference):

Reviewer's Responses to Questions

**Comments to the Author**

Reviewer #3: All comments have been addressed

Reviewer #4: (No Response)

publication criteria?

Reviewer #3: Yes

Reviewer #4: Yes

3. Has the statistical analysis been performed appropriately and rigorously?

Reviewer #3: Yes

Reviewer #4: Yes

4. Have the authors made all data underlying the findings in their manuscript fully available (please refer to the Data Availability Statement at the start of the manuscript PDF file)?

Reviewer #3: Yes

Reviewer #4: Yes

5. Is the manuscript presented in an intelligible fashion and written in standard English?

Reviewer #3: Yes

Reviewer #4: Yes

Reviewer #3: The document presented responds to all the very careful and comprehensive comments that were issued by previous reviewers. I should also add that, although a previous reviewer expressed that these findings were of local interest and hence, suited for a more regional journal, I disagree. These findings are useful for a variety of areas across the globe, where Leptospirosis is ubiquituous. More so with flooding and exteme weather events due to climate change.

Reviewer #4: According to the manuscript this is the first sex-disaggregated study on Leptospira infection risk, gender-differentiated perceptions, behaviours, and exposures in a high-burden settlement were explored. This is an important exploration intended to address knowledge gap in the field.

The authors claim was in line with related previous studies, the literature linkage is good enough as the systematic review and other studies cited lend credence to their work.

The cross-sectional design involving four Brazilian informal settlements with about 761 participant gives a good sample power. The reported use of standardized questionnaires that has been validated and Microscopic Agglutination Test (MAT) for serology as the gold-standard and other technique highlighted makes the study a sufficiently reproducible.

STROBE guidelines was used in the study which align with PLOS suggested relevant guidelines

The data presented supported the findings and discussion. Supplementary information was also given to clarify the regression analysis employed.

In the discussion segment, the authors explained the importance of conducting sex-disaggregated or gender-interaction analyses to identify gender-specific relationships that are obscured in combined model to buttress highlights of their findings.

Ethical approval and informed consent were gotten for the study as highlighted in this manuscript, so no ethical issue of concern.

The manuscript was well organized, written clearly enough as a good read for understanding of non-specialist except for some typographical error:- line 66 – spillover

line 107 – “a” written twice in the sentence

This paper can be considered for publication

**Do you want your identity to be public for this peer review?** For information about this choice, including consent withdrawal, please see our Privacy Policy

Reviewer #3: No

Reviewer #4: No
